# Role of Renin-Angiotensin-Aldosterone System and Cortisol in Endometriosis: A Preliminary Report

**DOI:** 10.3390/ijms24010310

**Published:** 2022-12-24

**Authors:** Chiara Sabbadin, Carlo Saccardi, Alessandra Andrisani, Amerigo Vitagliano, Loris Marin, Eugenio Ragazzi, Luciana Bordin, Guido Ambrosini, Decio Armanini

**Affiliations:** 1Endocrinology Unit, Department of Medicine, University of Padova, 35128 Padova, Italy; 2Department of Women’s and Children’s Health, University of Padova, 35128 Padova, Italy; 3Department of Pharmaceutical and Pharmacological Sciences, University of Padova, 35131 Padova, Italy; 4Department of Molecular Medicine-Biological Chemistry, University of Padova, 35131 Padova, Italy

**Keywords:** endometriosis, inflammation, hypertension, aldosterone, cortisol, mineralocorticoid receptor, spironolactone

## Abstract

Endometriosis is a chronic inflammatory disease associated with pelvic pain, infertility, and increased cardiovascular risk. Recent studies suggest a possible role of aldosterone as a pro-inflammatory hormone in the pathogenesis of the disease. Cortisol is also an important mediator of stress reaction, but its role is controversial in endometriosis. The aim of this study was to evaluate aldosterone and cortisol levels and blood pressure values in women with endometriosis. We measured blood pressure, plasma aldosterone, renin, cortisol, and dehydroepiandrosterone sulfate (DHEAS) in 20 women with untreated minimal or mild pelvic endometriosis compared with 20 healthy controls matched for age and body mass index. Aldosterone values were similar in the two groups, while renin was significantly lower and the aldosterone to renin ratio was significantly higher in patients with endometriosis than in controls. Systolic blood pressure was in the normal range, but significantly higher in patients with endometriosis. Morning plasma cortisol was normal, but significantly lower in patients with endometriosis compared with controls, while DHEAS to cortisol ratio was similar in the two groups. These preliminary results are evidence of increased biological aldosterone activity and dysregulation of the hypothalamic-pituitary-adrenal axis in early stages of endometriosis. These alterations could play a role in disease development, suggesting new therapeutic targets for aldosterone receptor blockers.

## 1. Introduction

Endometriosis is a chronic gynecological disease affecting 10% of women of reproductive age [1]. It is an inflammatory, estrogen-dependent disease characterized by the presence of endometrial foci outside the uterus, especially in the pelvic peritoneum and ovaries. This condition is often associated with chronic pelvic pain, dysmenorrhea, dyspareunia, infertility, and a reduced quality of life [2]. Some patients experience debilitating endometriosis-associated pain that prevents them from usual activities. Chronic inflammation plays a central role in the pathogenesis of atherosclerosis, obesity, diabetes, and hypertension [3], which are often associated with endometriosis [4]. In particular, an increased risk of occurrence of hypertension, gestational hypertension and pre-eclampsia has been reported among women with endometriosis [3,4,5,6]. Several mechanisms, including pro-inflammatory, pro-angiogenic, and aberrant immune-endocrine function, involved in the pathobiology of endometriosis demonstrate compelling associations with cardiovascular disease development. Other important factors may be impaired autophagy and mitophagy pathways and the consequent effect on apoptosis and angiogenesis [7]. Finally, even the alterations of sex hormones due to the pharmacological and surgical treatments could affect the risk of cardiovascular disorders.

In recent years, aldosterone has been considered an important pro-inflammatory factor responsible for increased cardiovascular risk, independent of blood pressure values [8]. Previous studies have reported mineralocorticoid receptor (MR) expression in non-classical targets of aldosterone, such as heart (cardiomyocytes, fibroblasts, vessels), endothelial cells, smooth muscle cells, and inflammatory cells [9,10]. Incubation of mononuclear leukocytes (MNL) with aldosterone induces the expression of markers of inflammation and oxidative stress, and this effect was reversed by coincubation with MR blocker canrenone [11]. Further studies have supported the role of aldosterone in the inflammatory and autoimmune mechanisms underlying several diseases [12]. In particular, aldosterone has also been involved in some gynecological conditions characterized by inflammation and increased cardio-metabolic risk, such as polycystic ovary syndrome (PCOS), preeclampsia, uterine fibroids, and endometriosis [13]. A significant increase in aldosterone or aldosterone to renin ratio (ARR) has been found in PCOS [14] and preeclampsia [15], actively contributing to the inflammatory milieu of these diseases. ARR has also been evaluated in uterine leiomyoma, showing a complex regulation of angiotensin II and aldosterone in the induction of leiomyoma cell proliferation [16,17]. A recent metabolomics-based study by Ghazi showed that aldosterone activity is higher in infertile patients with endometriosis [18]. Aldosterone could potentiate the systemic and local inflammation underlying endometriosis through activation of MR present in peritoneal and tissue inflammatory cells, particularly macrophages and MNL. Angiotensin II type-1 receptor and aldosterone synthase are specifically expressed in endometrial gland during mid-secretory phase. MR concentration is elevated in stroma in mid-secretory endometrium. In vitro, MR is also activated by aldosterone during decidualization [19]. However, the assessment of the effector mechanism of aldosterone is complex, and currently the ARR is considered the gold standard for the diagnosis of primary aldosteronism (PA), where even normal aldosterone concentrations may be associated with increased ARR [20]. In addition to aldosterone, glucocorticoids are also involved in the regulation of inflammatory processes. Cortisol at physiological concentrations exerts an anti-inflammatory activity, thus counteracting the pro-inflammatory action of aldosterone. Cortisol is the main mediator of the stress reaction, and its effect also seems to be regulated by the dehydroepiandrosterone sulfate (DHEAS), especially in the brain, where dehydroepiandrosterone (DHEA) and DHEAS are protective against possible brain damage due to cortisol excess [21,22]. The relationship between stress and endometriosis is not clear: previous studies on animal models have reported that stress increases the size and severity of the lesions as well as inflammatory parameters [23]. Under acute stress conditions, cortisol secretion increases to maintain body homeostasis by increasing heart rate, blood pressure, and glucose metabolism. However, after chronic exposure to stress, adverse health effects may occur due to dysregulation of the hypothalamic-pituitary-adrenal (HPA) axis, leading to alterations in cardiovascular and reproductive function, metabolism, psychological state, and immune-mediated inflammation [24]. The crosstalk between stress-inflammation-pain through HPA axis activity indicates that stress relief should alleviate inflammation and decrease painful responses [23]. Only a few studies have investigated the role of the HPA axis in patients with endometriosis, showing conflicting data [25,26]. The main purpose of this study was to evaluate aldosterone and cortisol levels and blood pressure values in women with endometriosis.

## 2. Results

Table 1 shows the anthropometric and biochemical parameters evaluated in women with endometriosis and healthy controls. Mean systolic and diastolic blood pressures were normal in all women, but systolic values were significantly higher in patients than controls (126.8 ± 8.2 vs. 117.1 ± 3.9 mmHg, *p* = 0.0003) (Figure 1). Plasma aldosterone concentration was higher in patients with endometriosis than controls, although not significantly different (Figure 2). On the contrary, renin levels were significantly lower in patients with endometriosis than in controls (15.9 ± 7.6 vs. 23.4 ± 8.3 mIU/L, *p* = 0.0055) (Figure 2). The ARR was normal in all enrolled women, excluding the diagnosis of PA, but significantly higher in patients with endometriosis than controls (28.1 ± 16.9 vs. 14.1 ± 6.1, *p* = 0.0001) (Figure 2).

Cortisol levels were normal in all women, but significantly lower in patients with endometriosis than controls (0.257 ± 0.051 vs. 0.399 ± 0.065 µmol/L, *p* < 0.0001) (Figure 2). DHEAS levels and DHEAS to cortisol ratio were similar in the two groups.

## 3. Discussion

Endometriosis is a complex chronic inflammatory condition associated with increased cardiovascular risk. Excess of aldosterone or of MR activity is known to induce inflammation and an increased cardiovascular risk. In contrast, cortisol is an anti-inflammatory hormone, released under stressful conditions. The involvement of these steroids in endometriosis is underestimated but could have important implications not only for better understanding of some of the physiopathological mechanisms of the disease, but also for considering new therapeutic targets. Therefore, the aim of this study was to evaluate plasma levels of aldosterone and cortisol and blood pressure values in patients with endometriosis, also exploring the ARR and DHEAS to cortisol ratio as indicators of aldosterone and cortisol activity, respectively. 

Our results show that plasma aldosterone concentrations were increased at the limit of significance (*p* = 0.0548) in patients with endometriosis, while renin levels were significantly reduced and the ARR was significantly increased compared with healthy controls. Indeed, in our sample of patients, systolic blood pressure levels were significantly higher than in healthy women. Considering that the ARR remains in the normal range, these results are consistent with a possible increased biological activity of aldosterone. We hypothesize that the increased MR activity in these patients may be related to MR hypersensitivity to aldosterone, which induces sodium retention, slight increase in systolic blood pressure, and decrease in renin levels. This feature is also common in normotensive and hypertensive black subjects [27] and in patients with essential low renin hypertension, who usually have good control of hypertension and reduced cardiovascular risk when treated with MR blockers [28,29]. It is difficult to establish the normal range of aldosterone, and ARR is the gold standard for the diagnosis of PA, plasma aldosterone often being in the normal range and some cases of PA have also been reported in normotensive subjects [30].

Ghazi and collaborators [18] hypothesized that metabolic profiles of patients with endometriosis are changed compared with healthy controls using Proton Nuclear Magnetic Resonance. The authors report an increased concentration of aldosterone metabolites in infertile women with endometriosis. The metabolomic approach enabled the identification of several metabolic alterations occurring in women with endometriosis. However, the author did not discuss the possible implications of the increased concentration of aldosterone metabolites. Our study is the first to highlight a possible involvement of aldosterone in this disease associated with an inflammatory state.

PA is associated with increased lymphocyte-mediated systemic inflammation and with increased oxidative stress in erythrocytes [31]. Both these effects are reversed by in vitro co-incubation of lymphocytes or erythrocytes with canrenone, hypothesizing that the genomic and non-genomic effects of aldosterone are mediated by classical MR. We also found a similar effect on erythrocytes of patients with endometriosis [32]. It is known that women with endometriosis have a higher risk of hypertension than healthy controls [33]. The coexistence of inflammatory status, increased MR and predisposition to hypertension suggest an involvement of the aldosterone effector mechanism as a concause or consequence of this disease and the associated increased cardiovascular risk.

A possible genetic and epigenetic role has been reported in the development of deep infiltrating endometriosis. The role of genetics is well established, with about 50% of the risk due to genetic factors and 50% due to environmental or other causes. Most somatic mutations are related to inflammation [34] and could be linked to an increased aldosterone effector mechanism, mediated by the presence of MR in both endometrial endothelial cells [19] and inflammatory cells infiltrating the endometrium. Aldosterone might play a role in disease progression, and MR antagonists might be useful in reducing endometriosis progression, inflammation, and cardiovascular risk.

Spironolactone (SP) is an aldosterone receptor blocker that is useful in preventing cardiovascular risk in patients with PA and resistant hypertension with normal aldosterone values [9]. In the last decade, considering all the benefits of MR antagonists, SP has been proposed for the treatment of many clinical situations, characterized by inflammation and increased cardiovascular risk. In particular, in PCOS, which is frequently associated with endometriosis, the use of SP has been proposed not only as an anti-androgen, but also as an anti-MR, to reduce the side effects related to the activation of the renin-angiotensin-aldosterone system, represented by increased ARR [14,35,36]. Because its chronic use is safe [37] and considering its effect in reducing fibrosis and intraperitoneal inflammation [38], SP could have therapeutic potential in endometriosis as monotherapy or associated with hormonal contraceptives or dienogest to treat hypertension and to prevent the progression of this disease.

In this study, we also evaluated cortisol and DHEAS levels and their possible role in patients with endometriosis. Our results show that patients with endometriosis have significantly lower morning cortisol levels than healthy women, with no differences in DHEAS and DHEAS to cortisol ratio. Our results are in agreement with previous studies that reported a relative hypocortisolism as a biomarker of aberrant HPA responses in endometriosis [26,39]. In particular, a study reported decreased salivary cortisol levels in several samples collected during the day, showing a significantly lower area under the curve in patients with endometriosis compared with healthy controls [26]. A possible role in the decrease of cortisol could be played by the presence of 11beta-hydroxysteroid dehydrogenase in endometrial cells and perhaps endometrial tissue, which partially inactivates cortisol to cortisone [40]. This relative hypocortisolism could be an adaptive response to the chronic stress associated with endometriosis and may be involved in the disease progression, as lack of cortisol availability may promote increased vulnerability to autoimmune disorders, inflammation, and chronic pain [41]. The implications of reduced cortisol levels are even more dangerous considering the increased effector mechanism of aldosterone in endometriosis and its possible role as a pro-inflammatory marker. However, other studies have found increased cortisol levels in patients with endometriosis, as an appropriate response to the emotional and physical stress induced by the disease [25,42]. Additionally, in the study by Ghazi using metabolomics, DHEA and androstenedione were found to be increased in patients with advanced endometriosis [18]. All our patients had untreated minimal or mild pelvic endometriosis: low stages could be associated with different HPA axis activation compared with advanced stages of endometriosis. Even previous treatments could influence HPA responses in affected patients. All of the above studies, however, suggest dysregulation of the HPA axis, caused by chronic stress resulting from endometriosis itself. Further studies should investigate whether alterations in the HPA axis are influenced by the duration, severity, and treatments of the disease, and whether they are involved in the disease progression.

Our study has some limitations: the small sample of patients, with only minimal or mild pelvic endometriosis; the lack of other signs related to cardiovascular risk; and the failure to measure other inflammatory markers. These preliminary observations will serve as a basis for future research with larger samples, also focusing on advanced endometriosis.

## 4. Materials and Methods

Twenty subjects with newly diagnosed minimal or mild pelvic endometriosis were enrolled at the Endoscopic Units of Gynecological Clinic of Padua (Department of Women’s and Children’s Health, University of Padua). Inclusion criteria were: age ranging from 18 to 40 years, stage I and II of endometriosis according to Revised American Society for Reproductive Medicine Classification of Endometriosis without previous specific treatments [43]. Exclusion criteria were: non-Caucasian ethnicity, systemic diseases which potentially interfere with inflammatory status (such as diabetes, asthma, thyroid and reumathologic disorders), history of hypertension, intake of estro-progestinic hormonal treatment or glucocorticoids in the previous three months. Twenty healthy women, matched for age and body mass index (BMI), were studied as a control group. PCOS was excluded by history, ultrasound analysis, and assessment of plasma androgen levels. Thyroid function was normal in all subjects studied. The study was approved by the Ethics Committee of the institutional authorities of the Azienda Ospedaliera-Universitaria di Padova, and the patients were fully informed of the purpose of the study and gave informed consent.

In all patients and controls, weight and height were collected, to calculate BMI (kg/m^2^), and systolic and diastolic blood pressure levels were measured twice in a sitting position by the same operator. For measurement of plasma aldosterone, renin, cortisol, and DHEAS, blood samples were collected from patients in the early follicular phase of their menstrual cycle. Blood samples were collected in the morning (between 7:30 and 8:30 a.m.), after the patient had been in an upright position for at least 30 min and then seated for 5 min during blood collection. The women were not taking contraceptives or other medications that could influence the hormonal assessment. Pregnancy was excluded by measurement of serum betaHCG.

All biochemical measurements were performed in the central laboratory of the Azienda Ospedaliera-Università di Padova. Plasma aldosterone and renin were measured by chemiluminescent assay (LIAISON^®^ XL Aldosterone kit and LIAISON^®^ Direct Renin kit, Diasorin, Saluggia, Italy). Serum cortisol and DHEAS were measured by electrochemiluminescence immunoassay (Roche).

### Statistical Analysis

Power analysis—Since no published data were available in the clinical condition of endometriosis, the evaluation of sample size was necessarily based on data pertaining to other gynaecological pathology. According to aldosterone plasma data from patients affected by polycystic ovary syndrome [14], the estimated effect size (Cohen’s d) in comparison to controls was d = 1.26, corresponding to a large effect [44]. Considering a prudential value of effect size d = 1, with a power (1 − β) of 0.80 and an α-error of 0.05, the theoretical sample size is estimated as 18 patients per group, which was set to 20 patients per group, due to the still unknown extent of the expected effect.

Data are expressed as mean ± SD. Normality of data distribution was evaluated with Shapiro–Wilk test. Since for several parameters a significant deviation from normality was detected, differences between groups were assessed by the nonparametric Mann–Whitney test, which is also suggested for analysis of small data sets [45]. A *p* value < 0.05 was considered statistically significant.

## 5. Conclusions

In conclusion, endometriosis is a complex condition characterized by chronic inflammatory state and increased cardiovascular risk. These features have also been observed in some other gynecological conditions, such as PCOS, and associated with a relative hyperaldosteronism. Our preliminary results suggest a possible role of increased biological aldosterone activity in patients with endometriosis. An important role of reduced cortisol levels is possible in this disease, at least in the early stages. For these reasons, we believe that the use of SP may have important implications in the treatment and prevention of endometriosis. Further studies are needed to better explore this new aspect of the pathogenesis of endometriosis.

## Figures and Tables

**Figure 1 ijms-24-00310-f001:**
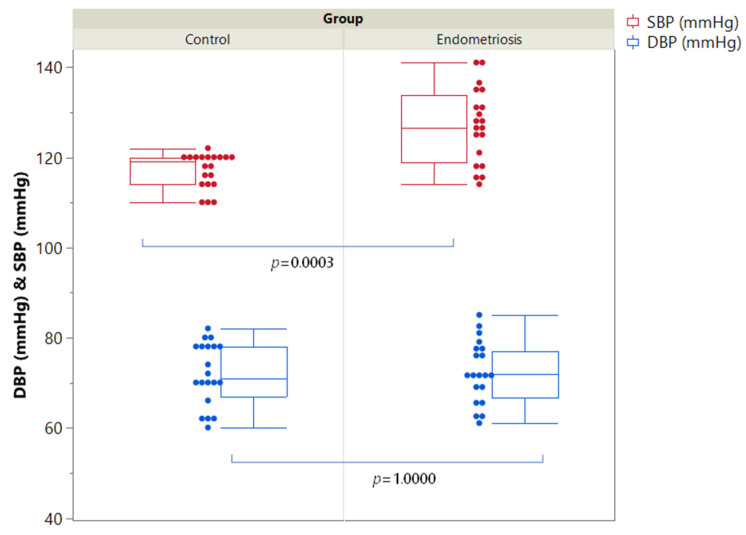
Distribution of Systolic (SBP) and Diastolic (DBP) blood pressure in the two groups of women. Single data points and box plots are presented. The edges of the boxes indicate the 25th and 75th quantiles, including the middle 50 percent of the data; the whiskers represent the range of data, calculated as (upper-quartile + 1.5×IQR) and (lower quartile − 1.5×IQR).

**Figure 2 ijms-24-00310-f002:**
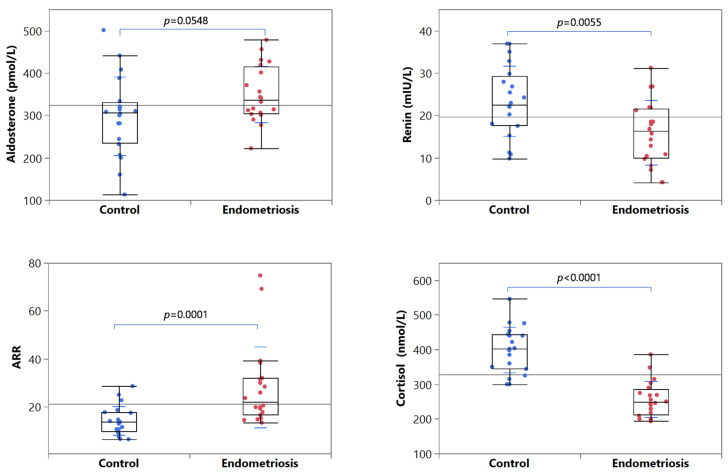
Distribution of aldosterone, renin, ARR, and cortisol values among healthy controls and patients with endometriosis. The edges of the boxes indicate the 25th and 75th quantiles, including the middle 50 percent of the data; the whiskers represent the range of data, calculated as (upper quartile + 1.5×IQR) and (lower quartile − 1.5×IQR). The continuous horizontal line is the overall arithmetic mean of the whole data set.

**Table 1 ijms-24-00310-t001:** Anthropometric and biochemical parameters evaluated in patients with endometriosis and healthy controls.

Parameters	Endometriosis (*n* = 20)	Control (*n* = 20)	*p*
Age (y)	34.3 ± 4.7	34.0 ± 3.0	0.4544
BMI (kg/m^2^)	20.0 ± 1.5	20.7 ± 1.3	0.1937
SBP (mmHg)	126.8 ± 8.2	117.1 ± 3.9	0.0003
DBP (mmHg)	72.3 ± 6.9	72.0 ± 6.9	1.0000
Aldosterone (pmol/L)	349.8 ± 66.6	298.3 ± 92.9	0.0548
Renin (mIU/L)	15.9 ± 7.6	23.4 ± 8.3	0.0055
ARR	28.1 ± 16.9	14.1 ± 6.1	0.0001
Cortisol (µmol/L)	0.257 ± 0.051	0.399 ± 0.065	<0.0001
DHEAS (µmol/L)	3.5 ± 1.8	4.4 ± 1.6	0.1228
DHEAS/cortisol	13.7 ± 6.7	11.1 ± 5.8	0.1988

BMI: Body Mass Index; SBP: Systolic blood pressure; DBP: Diastolic blood pressure; ARR: Aldosterone to renin ratio.

## Data Availability

The data presented in this study are available on request from the corresponding author.

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
