# Peer review of "Role of Renin-Angiotensin-Aldosterone System and Cortisol in Endometriosis: A Preliminary Report"

_ijms, 2022, doi:10.3390/ijms24010310_

Round 1
Reviewer 1 Report
Dear Authors,
The manuscript is interesting, and as a continuation of Your research regarding RAA/cortisol studies in various clinical conditions. The study is interesting, however it requires some major changes.
In abstract, You present the aims of Your study. One of the aims is to correlate various hormone levels with blood pressure. I do not understand why this vital sign is so important (and not for example heart rate and/or respiratory rate). You have to explain it to me in introduction section. It seems to be one of the main aims of Your research and You do not mention anything about coincidence of increased blood pressure values in patients with endometriosis - the introduction section has to be corrected. Please add some information regarding the incidence of increased blood pressure in endometriotic patients, why it is important in Your opinion, add some information about coincidence of other vital signs abnormalities and endometriosis. Next, in discussion section You write: "... the aim of this study was to evaluate plasma aldosterone and cortisol levels in patients with endometriosis, exploring also the ARR and DHEAS/cortisol ratio as indicators of aldosterone and cortisol activity, respectively..." So what about blood pressure? Please correct. Noteworthy, You discuss blood pressure changes in patients with endometriosis, so it is important part of Your research. Again, some information regarding other vital signs changes in endometriosis should be added in this section.
Although it is a communication manuscript, it requires some more information in methodology section. How the levels of hormones were assessed? What kind of tests were used: immunoassays? others? Please add manufacturer. When were the tests performed in relation to diagnostic/therapeutic endometriosis surgery? A cortisol may be increased in stress situation and surgery is undoubtedly such a condition, the results may not be reliable and cannot be extrapolated to potential chronic treatment of a chronic disease. If the blood tests were taken perioperatively You have to describe it in methodology section and try to discuss these results (in discussion section), trying to answer the question: How the stressful situation (in this case surgery) could change the concentrations of those hormones?
Lastly, please carefully read the whole manuscript to find and correct all typing/grammatical errors. Some of them are listed below:
phisyopathological-line 108;
concause-line 142;
... the possible implications of the increase concentration... - line 132;
He reports - line 129 (Dr Ghazi is female - if You do not know the author it is always safe to write the authors...),
.... significantly higher in patients. - line 20 (endometriotic patient/patients with endometriosis?)
endometriosis - line 21 - (endometriotic patients?)
mononuclear leukocute - line 43
due to excess cortisol-line 67 - due to cortisol excess
metabo-lism line 69
erithrocytes - line 139
Author Response
The manuscript is interesting, and as a continuation of Your research regarding RAA/cortisol studies in various clinical conditions. The study is interesting, however it requires some major changes.
- Thanks for your suggestions that improved the paper.
In abstract, You present the aims of Your study. One of the aims is to correlate various hormone levels with blood pressure. I do not understand why this vital sign is so important (and not for example heart rate and/or respiratory rate). You have to explain it to me in introduction section. It seems to be one of the main aims of Your research and You do not mention anything about coincidence of increased blood pressure values in patients with endometriosis - the introduction section has to be corrected. Please add some information regarding the incidence of increased blood pressure in endometriotic patients, why it is important in Your opinion, add some information about coincidence of other vital signs abnormalities and endometriosis. Next, in discussion section You write: "... the aim of this study was to evaluate plasma aldosterone and cortisol levels in patients with endometriosis, exploring also the ARR and DHEAS/cortisol ratio as indicators of aldosterone and cortisol activity, respectively..." So what about blood pressure? Please correct. Noteworthy, You discuss blood pressure changes in patients with endometriosis, so it is important part of Your research. Again, some information regarding other vital signs changes in endometriosis should be added in this section.
- We have improved the introduction adding more detalis e references related to the increased association between endometriosis and hypertension. Unfortunately, we did not measure heart rate and other vital signs in the sample of enrolled patients.
Although it is a communication manuscript, it requires some more information in methodology section. How the levels of hormones were assessed? What kind of tests were used: immunoassays? others? Please add manufacturer. When were the tests performed in relation to diagnostic/therapeutic endometriosis surgery? A cortisol may be increased in stress situation and surgery is undoubtedly such a condition, the results may not be reliable and cannot be extrapolated to potential chronic treatment of a chronic disease. If the blood tests were taken perioperatively You have to describe it in methodology section and try to discuss these results (in discussion section), trying to answer the question: How the stressful situation (in this case surgery) could change the concentrations of those hormones?
- We have added the requested details in material and methods. We have also discussed the possible implications of disease stage and other stressful events on cortisol levels.
Lastly, please carefully read the whole manuscript to find and correct all typing/grammatical errors. Some of them are listed below:
phisyopathological-line 108;
concause-line 142;
... the possible implications of the increase concentration... - line 132;
He reports - line 129 (Dr Ghazi is female - if You do not know the author it is always safe to write the authors...),
.... significantly higher in patients. - line 20 (endometriotic patient/patients with endometriosis?)
endometriosis - line 21 - (endometriotic patients?)
mononuclear leukocute - line 43
due to excess cortisol-line 67 - due to cortisol excess
metabo-lism line 69
erithrocytes - line 139
- Thank you, we have revised the English grammar.
Reviewer 2 Report
The authors have presented an adequate study, well written and with innovative and quality data. However, it is necessary for the authors to make minor changes.
-First, the authors should improve the quality of Figure 1.
-The authors must improve the figure legends.
-The authors should better explain the statistical section.
-The authors must adequately justify the inclusion and exclusion criteria. As well as the sample size.
-The authors should improve the use of English grammar.
Author Response
The authors have presented an adequate study, well written and with innovative and quality data. However, it is necessary for the authors to make minor changes.
- We would like to thank for your revision suggestions that have improved our paper.
First, the authors should improve the quality of Figure 1.
- Figure 1 quality has been improved, including also another parameter (renin) for a better visual appreciation of the overall results.
The authors must improve the figure legends.
- We improved the figure legend, providing information also regarding the box plot.
The authors should better explain the statistical section.
- A more detailed statistical section has been provided.
The authors must adequately justify the inclusion and exclusion criteria. As well as the sample size.
- Done.
The authors should improve the use of English grammar.
- Done.
Reviewer 3 Report
The authors investigated about the role of aldosterone and cortisol levels in women with endometriosis. The manuscript is almost well written. Overall the topic could be interesting but some details could be improved.
I recommend that the paper be accepted with minor revision:
a) In the introduction section, little previous evidence is provided about the importance of endometriosis in daily life. Incorporating comparisons with other studies would increase the strength of the paper. Please refer to doi: 10.1007/s43032-020-00205-7; 10.3390/ijms22105074; 10.1038/s41419-020-02844-9; 10.18632/oncotarget.25823.
b) The authors should better emphasize the conclusions.
c) There are some minor grammar issues that should be fixed in order to aid the accessibility of the results to the reader.Author Response
The authors investigated about the role of aldosterone and cortisol levels in women with endometriosis. The manuscript is almost well written. Overall the topic could be interesting but some details could be improved.
I recommend that the paper be accepted with minor revision:
a) In the introduction section, little previous evidence is provided about the importance of endometriosis in daily life. Incorporating comparisons with other studies would increase the strength of the paper. Please refer to doi: 10.1007/s43032-020-00205-7; 10.3390/ijms22105074; 1038/s41419-020-02844-9; 10.18632/oncotarget.25823.
- Thank you for your suggestions, that have improved our paper.
b)The authors should better emphasize the conclusions.
- Done.
c) There are some minor grammar issues that should be fixed in order to aid the accessibility of the results to the reader.
- The English grammar has been revised.
Reviewer 4 Report
The proposed article "Role of renin-angiotensin-aldosterone system and cortisol in endometriosis" presents the metabolic profile of endometriosis. Chronic inflammation is associated with cardiovascular diseases. The role of aldosterone and cortisol and their negative effect on the hypothalamic pituitary adrenal axis seems to be a very interesting topic and gives us a different insight into endometriosis origin. The change in aldosterone and cortisol levels in patients with endometriosis could give us a different perspective on the prevention and treatment of such a chronic condition. The main purpose of the article was to evaluate the aldosterone and cortisol level in patients with endometriosis. However, in the Discussion, the authors said that the hypothesis is that the increased MR activity in these patients may be related to MR hypersensitivity to aldosterone, which induces sodium retention, a slight increase in systolic blood pressure, and a decrease in renin levels. It is important to clarify the research question and possibly to include other parameters in the Title such as renin and sodium level. The presentation of the Results is good, however very small and not enough. So, my suggestion is to present more results in graphic form. Maybe you could include sodium level as well as other inflammation markers such as C - reactive protein or others in order to explain the risk of cardiovascular risk in patients with endometriosis. It is a preliminary report, and it should be stated in the Title. I agree with the authors that the major limitations of the study are the small sample size and the inclusion of patients with minimal or mild pelvic endometriosis. I strongly recommend including patients with advanced endometriosis. Statistical analysis is appropriate but I could not see it in total maybe due to technical problems. There are no ethical concerns or other issues I believe the Editor should be aware of.
These are my guide to authors on how they can strengthen their manuscript to the point where it might be acceptable for publication.
Author Response
The proposed article "Role of renin-angiotensin-aldosterone system and cortisol in endometriosis" presents the metabolic profile of endometriosis. Chronic inflammation is associated with cardiovascular diseases. The role of aldosterone and cortisol and their negative effect on the hypothalamic pituitary adrenal axis seems to be a very interesting topic and gives us a different insight into endometriosis origin. The change in aldosterone and cortisol levels in patients with endometriosis could give us a different perspective on the prevention and treatment of such a chronic condition. The main purpose of the article was to evaluate the aldosterone and cortisol level in patients with endometriosis. However, in the Discussion, the authors said that the hypothesis is that the increased MR activity in these patients may be related to MR hypersensitivity to aldosterone, which induces sodium retention, a slight increase in systolic blood pressure, and a decrease in renin levels. It is important to clarify the research question and possibly to include other parameters in the Title such as renin and sodium level. It is a preliminary report, and it should be stated in the Title.
- thank you for your comments. We have changed the title of our communication.
The presentation of the Results is good, however very small and not enough. So, my suggestion is to present more results in graphic form. Maybe you could include sodium level as well as other inflammation markers such as C - reactive protein or others in order to explain the risk of cardiovascular risk in patients with endometriosis.
- thanks, we improved the graphical appearance of the figure, including also another parameter (renin) for a better visual appreciation of the results. Regarding other suggested markers of inflammation, unfortunately they are not available for patients included in the present investigation. Indeed, we have added this in the limitations of our study. As suggested by the Reviewer, we will surely consider evaluating other inflammation markers in a future investigation.
I agree with the authors that the major limitations of the study are the small sample size and the inclusion of patients with minimal or mild pelvic endometriosis. I strongly recommend including patients with advanced endometriosis.
- we agree with you. These results are only preliminary and related to untreated patients with minimal or mild disease. Further studies should consider even patients with advanced endometriosis.
Statistical analysis is appropriate but I could not see it in total maybe due to technical problems. There are no ethical concerns or other issues I believe the Editor should be aware of.
- A more detailed statistical section has been provided, including sample size determination.
These are my guide to authors on how they can strengthen their manuscript to the point where it might be acceptable for publication.
- Thanks to your suggestions that have improved our paper.
Round 2
Reviewer 1 Report
The paper can be accepted for publication now.
Author Response
The manuscript by Sabbadin et al. presents data based on human patients, making these results of considerable value. However, a few changes need to be made before publication of the manuscript.
1. The abbreviations DHEA and DHEAS need to be defined at first use.
Done.
2. The term "direct renin" is used in the abstract but not in the manuscript text. This can be confusing to the general reader. One possibility is to remove "direct" from the abstract: the interested reader can note that plasma renin was measured using a LIAISON Direct Renin kit. If the authors determine that "direct" should not be removed form the abstract, then at the first mention of "renin" in the Results section the authors need to explain that plasma renin was measured using a LIAISON Direct Renin kit which is distinct from kits that measure plasma renin activity.
Thank you for your careful annotation. We have simply removed the term “direct” from the abstract.
3. When determining the upright aldosterone/renin levels, the authors need to state the length of time that the patients were in an upright position before blood samples were taken.
Done.
Lines 154-155 state " and ARR is the gold standard for the diagnosis of PA...." Does this refer to upright ARR?
The answer is quite complicated, since the guidelines have conflicting positions. In particular, the Endocrine Society guidelines suggest to perform the ARR test in the morning after patients have been out of bed for at least 2 hours, usually after they have been seated for 5–15 minutes; on the contrary, the Italian Society of Arterial Hypertension guidelines recommend to perform blood samples after 60 min of quiet supine or sitting rest. In the clinical practice, as reported by a very recent meta-analysis by Zhu R. et al published in the J Clin Endocr Metab, there are many protocols to prepare the patient before the evaluation of ARR. For all this reason I think it is not useful to describe all these aspects in the text. What we know is that ARR is the best tool to detect PA; the upright position could increase renin levels and underestimate a PA; however we share the guidelines by Endocrine Society.
Some changes in English diction should also be made.
A) Lines 28-29
"These preliminary results evidence an increased...."
should be changed to
These preliminary results are evidence of increased...."
B) Lines 49-50
"Other important factors may be the impaired autophagy and mitophagy pathways on apoptosis and angiogenesis [7]."
should be changed to
"Other important factors may be impaired autophagy and mitophagy pathways [7]."
Note that impaired autophagy and mitophagy pathways likely affect more than apoptosis and angiogenesis. Also, the text does not discuss the affect of impaired autophagy and mitophagy pathways on apoptosis and angiogenesis.
C) Line 138
"better understanding some physiopathological mechanisms of the disease...."
should be changed to
"better understanding of some of the physiopathological mechanisms of the disease...."
D) Lines 143-145
"Our results show that plasma aldosterone concentrations are increased at the limit of significance (p=0.0548) in patients with endometriosis, while renin levels are significantly reduced and the ARR increased compared with healthy controls."
should be changed to
"Our results show that plasma aldosterone concentrations were increased at the limit of significance (p=0.0548) in patients with endometriosis, while renin levels were significantly reduced and the ARR was significantly increased compared with healthy controls."
E) Line 164
"disease associated with major inflammatory state."
should be changed to
"disease associated with an inflammatory state."
The state of inflammation, major, medium, mild, or minor is not defined in this manuscript.
F) Lines 168-169
"We also found a similar effect even on erythrocytes of patients with endometriosis [32]."
should be changed to
"We also found a similar effect on erythrocytes of patients with endometriosis [32]."
G) Line 172
"hypertension may suppose an involvement...."
should be changed to
"hypertension suggest an involvement...."
H) Lines 190-193
"Because its chronic use is safe [37], SP could have therapeutic potential in endometriosis as monotherapy or associated with hormonal contraceptives or dienogest to treat hypertension and to prevent the progression of this disease, considering its effect in reducing fibrosis and intraperitoneal inflammation [38]."
should be changed to
"Because its chronic use is safe [37] and considering its effect in reducing fibrosis and intraperitoneal inflammation [38], SP could have therapeutic potential in endometriosis as monotherapy or associated with hormonal contraceptives or dienogest to treat hypertension and to prevent the progression of this disease."
I) Lines 229-230
"Our preliminary results suggest a possible role of increased aldosterone activity even in patients with endometriosis."
should be changed to
"Our preliminary results suggest a possible role of increased aldosterone activity in patients with endometriosis."
J) Line 231
"cortisol levels is conceivable in this disease"
should be changed to
"cortisol levels is possible in this disease"
K) Lines 254-255
"Plasma aldosterone, renin, cortisol, and DHEAS were performed in the morning (between 7:30 and 8:30 a.m.), in the upright position, during the early follicular phase."
should be changed to something like
"For measurement of plasma aldosterone, renin, cortisol, and DHEAS, blood samples were collected from patients in the early follicular phase of their menstrual cycle. Blood samples were collected in the morning (between 7:30 and 8:30 a.m.), after the patient had been in an upright position for xx hr (or yy min)."
Thank you so much for all your corrections and suggestions that have improved our paper.